# Fast Ground Segmentation for 3D LiDAR Point Cloud Based on Jump-Convolution-Process

Zhihao Shen [1], Huawei Liang [2,3], Linglong Lin [2,4,*], Zhiling Wang [2,3], Weixin Huang [1,2] and Jie Yu [2,4]

1　School of Information Science and Technology, University of Science and Technology of China, Hefei 230026, China; szh127@mail.ustc.edu.cn (Z.S.); hwx2018@mail.ustc.edu.cn (W.H.)
2　Hefei Institutes of Physical Science, Chinese Academy of Sciences, Hefei 230031, China; hwliang@iim.ac.cn (H.L.); zlwang@hfcas.ac.cn (Z.W.); yujieahaq@rntek.cas.cn (J.Y.)
3　Anhui Engineering Laboratory for Intelligent Driving Technology and Application, Hefei 230031, China
4　Innovation Research Institute of Robotics and Intelligent Manufacturing, Chinese Academy of Sciences, Hefei 230031, China
*　Correspondence: linll@iim.ac.cn

**Abstract:** LiDAR occupies a vital position in self-driving as the advanced detection technology enables autonomous vehicles (AVs) to obtain much environmental information. Ground segmentation for LiDAR point cloud is a crucial procedure to ensure AVs' driving safety. However, some current algorithms suffer from embarrassments such as unavailability on complex terrains, excessive time and memory usage, and additional pre-training requirements. The Jump-Convolution-Process (JCP) is proposed to solve these issues. JCP converts the segmentation problem of the 3D point cloud into the smoothing problem of the 2D image and takes little time to improve the segmentation effect significantly. First, the point cloud marked by an improved local feature extraction algorithm is projected onto an RGB image. Then, the pixel value is initialized with the points' label and continuously updated according to image convolution. Finally, a jump operation is introduced in the convolution process to perform calculations only on the low-confidence points filtered by the credibility propagation algorithm, reducing the time cost. Experiments on three datasets show that our approach has a better segmentation accuracy and terrain adaptability than those of the three existing methods. Meanwhile, the average time for the proposed method to deal with one scan data of 64-beam and 128-beam LiDAR is only 8.61 ms and 15.62 ms, which fully meets the AVs' requirement for real-time performance.

**Keywords:** autonomous vehicles; LiDAR; ground segmentation; convolution; real-time





## 1. Introduction

LiDAR [1] is widely used in AVs due to its stability and accuracy compared with the easily disturbed camera by light and weather. It is essential to process the raw point cloud in the perception phase because what the planning system ultimately needs is an accessible area and the obstacles' location [2–4]. Ground segmentation is a material priority work in conventional perception tasks, which is the basis for Clustering, Recognition, and Tracking. Furthermore, the accuracy and the time delay of ground segmentation directly determine the safe driving speed of AVs. Less processing time enables the system to receive more LiDAR scan data, and higher accuracy can restore a more realistic surrounding environment. The vehicle's safety can be guaranteed by having both of them, which is a necessary condition for AVs to be able to drive at high speed [5].

Considering the diversity of the outdoor driving environments and the technical limitations of the mobile LiDAR, the primary difficulties faced by current researches on the ground segmentation of 3D point clouds are as follows: (1) AVs' encounters on rugged terrain concerning pitch, roll, and suspension changes, resulting in abnormal features of the point cloud. (2) The density distribution of the point cloud is not uniform, and points

far away from the LiDAR sensor are sparse and poorly characterized. (3) More than 100,000 points need to be handled for every scan. It is arduous for the onboard computer to achieve a balance with accuracy and speed.

At present, the ground segmentation of the LiDAR point cloud can be roughly divided into the following methods [6]: the method based on elevation, the method based on the relationship between adjacent points, the method based on road modeling, the method based on LiDAR image, the method based on Markov, and the method based on deep learning.

Researchers at Stanford University first proposed an elevation map method [7], which divided the points into ground and non-ground based on the relative height difference. This algorithm allowed their AVs to win the 2005 DARPA Grand Challenge, but simple elevation maps are unsuitable for complex scenes, so scholars have obtained better segmentation effects by adding postprocessing or changing the grid structure. Zermas et al. [8] extracted the road surface according to the initial seed ground points generated from a simple elevation map and performed iterative clustering to improve the segmentation effect. A multi-volume grid structure is introduced in [9] to increase the segmentation accuracy. The main strength of this structure is that it is capable of coping with protruding and hanging objects found in urban scenarios.

By strictly limiting the spatial distance and angle of the front and back points, the typical terrain can be segmented accurately and effectively [10–12]. Denoising and correcting the 3D point cloud would help to improve the segmentation effect [13]. Cheng et al. [14] analyzed the adjacent line segments' characteristics, marking them as the ground segments and obstacle segments based on the height, distance, and angle. In [15], the points on the same horizontal line are also filtered to improve the segmentation effect on complex terrains. However, the methods based on the relationship between adjacent points are poorly applicable because they usually need to set different parameters for different terrains.

Road modeling is an essential branch in the ground segmentation of the 3D point cloud. Although the angle information between the points could be used to simulated the object boundary through the region growing algorithm [16,17], this is only suitable for flat roads with apparent boundaries. Gaussian process regression (GPR) is widely adopted in this field [18–20]. Using the covariance function to establish a one-dimensional or two-dimensional Gaussian model could predict the road height accurately. Liu et al. [21] integrated the GPR and robust locally weighted regression (RLWR) by dividing the point cloud projected on the polar grid map into radial and circumferential filtering processes to form a hybrid regression model, eliminating the outliers' influence and predicting the ground surface robustly. However, these methods require obtaining accurate pavement seed points beforehand, which is a challenge for rugged roads.

Considering the fixed angle relationship between the LiDAR scanner rays, Bogoslavskyi et al. [22,23] proposed to map the point cloud into a depth image and used the neighborhood relationship to segment the depth image. This ingenious transformation gave later researchers a new direction [24,25]. Because of the flexibility of the graph structure, after mapping points into pixels, using a rich image processing library can quickly and accurately get a good segmentation effect. In [26], the flood-fill algorithm is applied to optimize the depth image segmentation result. Moreover, the clustering operation [27] is added after the first segmentation, which improves the effect significantly. Yang et al. [28] developed and proposed a Two-Layer-Graph structure on this basis, which described the point clouds hierarchically. They used a range graph to represent point clouds and a set graph for point cloud sets, reducing processing time and memory consumption.

Several of the road detection methods based on Markov Random Field (MRF) [29–31] have the potential for ground segmentation. They used gradient cues of road geometry to construct MRF and implemented belief propagation algorithms to classify the surrounding environment into different regions. Refs. [32,33] extended in this direction and proved that this method could accurately segment the point cloud even in undulating roads (such as downhill/uphill). Nevertheless, using belief propagation for reasoning about MRF is computationally expensive, which limits the prospect of real-time performance.

Huang et al. [34] initialized the MRF model by the coarse segmentation results obtained from the local feature extraction rather than prior knowledge, dramatically reducing modeling time. Then, the graph cut method is used to minimize the proposed model to achieve fine segmentation.

With the advent of PointNet [35,36], many researchers have set their sights on applying deep learning to three-dimensional point clouds [37,38]. GndNet [39] divided the point cloud into a small cell by separating the grid, then used PointNet and Pillar Feature Encoding network to extract features and regress ground height for each cell. In Cylinder3D [40], the cylinder partition and asymmetrical 3D convolution networks were designed to handle the inherent difficulties in the outdoor LiDAR point cloud, namely, sparsity and varying density, effectively and robustly. Although they have perfect segmentation effects on the Semantickitti dataset [41], the lack of datasets, the poor interpretability of the network, and the time-consuming conversion between 3D point cloud data and network input data are all issues that are hard to solve at the present stage and limit the development of convolutional neural networks. As a result, scholars still hold a wait-and-see attitude toward applying convolutional neural networks to the point cloud [42].

In this paper, we distribute the ground segmentation into two stages. We extract the local point cloud height characteristic and a ground characteristic to quickly get a rough segmentation result in the first stage. Then, we apply a method based on image convolution with low computational complexity to refine the segmentation result further in the second stage. Our contributions are summarized as follows:

- The ground height characteristic is attached to the elevation map, which breaks through its limitations in complex road scenes, and significantly improves the over-segmentation compared with the usual elevation map method.
- An image smoothing method using Jump-Convolution-Process (JCP) is proposed to optimize the segmentation effect. As a result, our method has a better segmentation effect, less processing time, and more robust terrain adaptability than other methods.

Two experimental results in the urban road environment and wild road environment are shown in Figure 1 (All drawings in this paper, the ground points are denoted as green, the obstacle points are denoted as red).

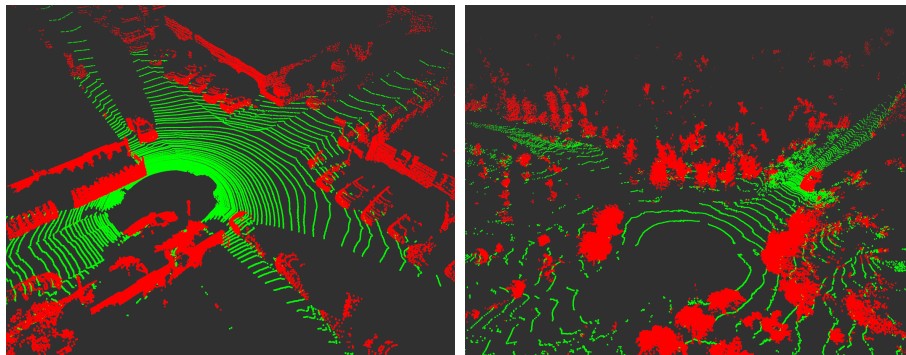

**Figure 1.** Illustration of the 3D point cloud segmentation applying JCP on the urban road (**left**) and the field road (**right**). Ground points are green; Obstacle points are red.

The remainder of this paper is organized as follows. Section 2 describes the proposed method specifically. Then, Section 3 shows and analyzes the experiment results, Section 4 carries out relevant discussions, and Section 5 summarizes the article and prospects for future work.

## 2. Methods

The proposed method is developed on the coarse-to-fine segmentation structure [8,27,28,34], accessing the entire point cloud as a single input and assigning each point to a category. The overall framework is shown in Figure 2. To facilitate data indexing, we sort the LiDAR data in

advance according to the angular relationships, detailed in Section 2.1. First, the raw point cloud is pre-classified as ground points or obstacle points by an improved feature extraction algorithm—the ring-shaped elevation conjunction map (RECM), which detailed in Section 2.2. Then, the point cloud is mapped to an RGB image, and the pixel value is initialized depending on the result of the coarse segmentation to generate the "Ground-Obstacle Image". Finally, the jump convolution process (JCP) is executed to re-classify the "low-confidence points"—the intersections between the green and red channels after credibility propagation, and optimize the segmentation result, detailed in Section 2.3.

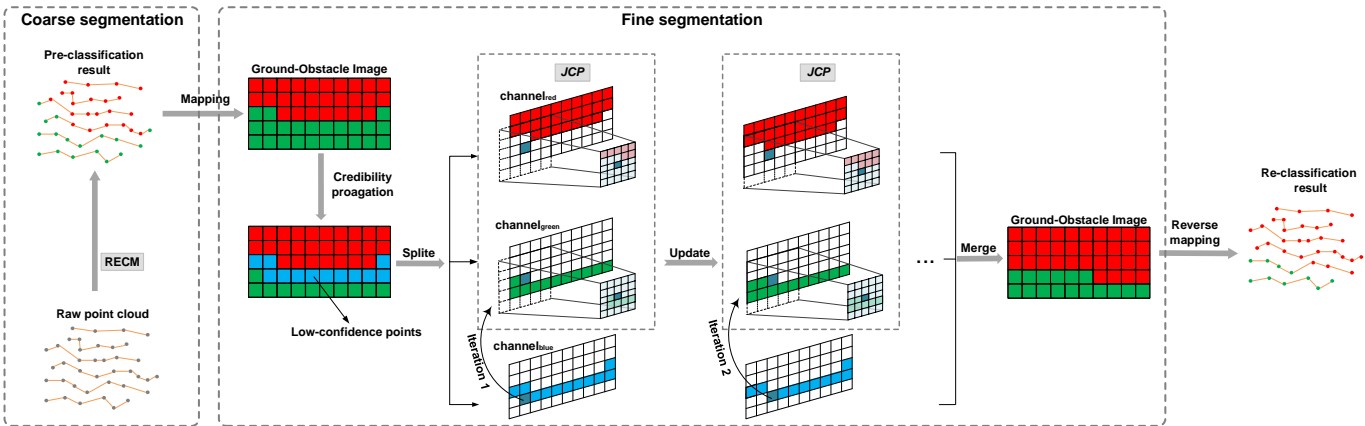

**Figure 2.** The framework of the proposed method. RECM and JCP are for ring-shaped elevation conjunction map and jump convolution process, respectively.

### 2.1. Point Cloud Sorting

To improve the algorithms' performance, we order the raw point cloud by employing the angular relationships between the laser beams. A LiDAR system contains three spherical coordinates: rotation angle $\beta$ (Figure 3a), vertical angle $\gamma$ (Figure 4a), and measurement distance $d$. Thus, we define a single LiDAR point as $p_{(t,c)}$, and its data structure $\{x_{(t,c)}, y_{(t,c)}, z_{(t,c)}, label_{(t,c)}\}$ is shown in Figure 3c. Among them, $(t, c)$ represents the serial number of this point; $x$, $y$, and $z$ represent the three-dimensional Cartesian coordinates; and *label* represents its type. The conversion relationship between the spherical coordinates and the Cartesian coordinate system is formulated as

$$\begin{cases} x = d \cdot \cos\gamma \cdot \sin\beta \\ y = d \cdot \cos\gamma \cdot \cos\beta \\ z = d \cdot \sin\gamma \end{cases} \tag{1}$$

the $\gamma$ of the same laser beam is constant, while the $\beta$ will be increased with a fixed horizontal angular resolution $\alpha$ (usually $0.18°$) over time. Therefore, the point cloud can be sorted by

$$\begin{cases} t = f(\gamma), & 0 \le t < T \\ c = \dfrac{\beta}{\alpha}, & 0 \le c < C \end{cases} \tag{2}$$

where $f(*)$ denotes the corresponding connection between the $\gamma$ and the laser beam sequence, and $t$ and $T$ are the serial number of the laser beam (arrange from bottom to up) and total numbers of the laser beams, respectively. $c$ is the serial number corresponding to the rotation angle (clockwise direction), and $C$ is the total number of points obtained by one laser beam scans $360°$.

$$C = \frac{360°}{\alpha} \tag{3}$$

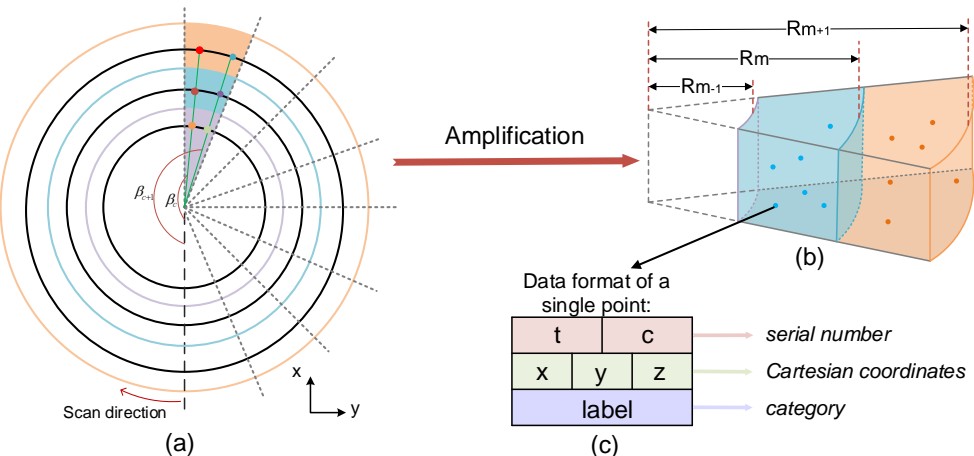

**Figure 3.** (**a**) The schematic diagram of the ring-shaped elevation map. The black circles represent the LiDAR scan's ideal shape, and different colored blocks represent different cells. (**b**) The three-dimensional display of the adjacent cells and the corresponding stored points. $R_m$ is the radius of the $grid_{(m,n)}$. (**c**) The data form of a single point. $t$, $c$ are the serial number; $x$, $y$, $z$ are the Cartesian coordinates; and *label* is the category.

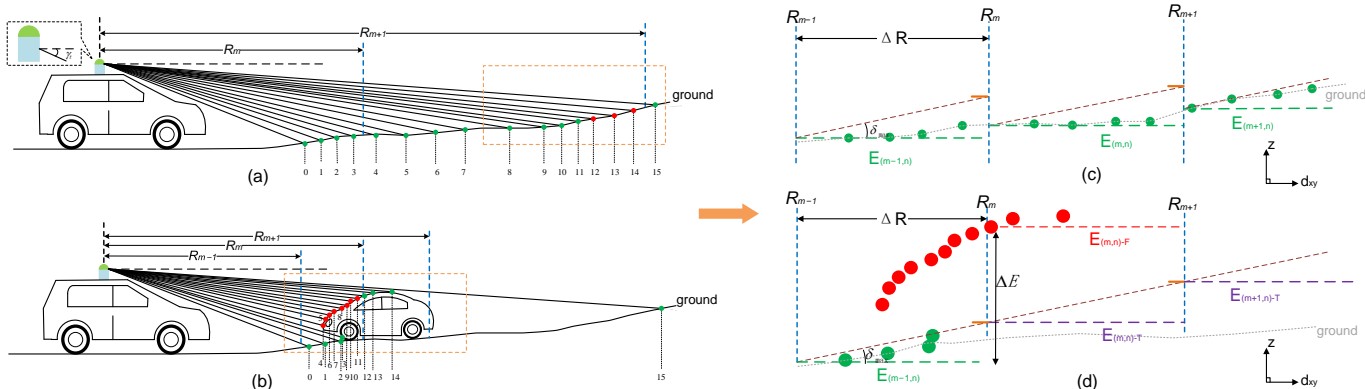

**Figure 4.** The segmentation results of different $\Delta R$ values and the segmentation principle diagram of the ring-shaped elevation conjunction map (RECM). (**a**) A large value of $\Delta R$ causes under-segmentation. The distant road surface is considered as an obstacle. (**b**) A small value of $\Delta R$ causes over-segmentation. The car's roof is regarded as the ground. (**c**) Points on the slope are corrected to the ground by the RECM. (**d**) Points on the roof are corrected to the obstacles by the RECM.

### 2.2. Ring-Shaped Elevation Conjunction Map

The elevation map has demonstrated its reliable performance in many methods [9,20]. As the laser beam scan roughly circle-shaped, it is more practical to establish a ring-shaped elevation map (REM) [19] to meet our requirements. We set up an elevation map $E$ of $M \times N$ size on the *xoy* plane, cutting the plane into $M - 1$ rings with LiDAR as the center of the circles; every ring is equally divided into $N$ parts. Figure 3a shows that each cell $grid_{(m,n)}$ is expressed by a colored block, representing the $n - th$ part of the $m - th$ ring. Provided the height is ignored, all points will be assigned into a grid related to their plane distance and rotation angle $\beta$. Namely, two conditions should be met if $p_{(t,c)} \in grid_{(m,n)}$:

$$
\begin{cases}
R_m \le \sqrt{x_{(t,c)}^2 + y_{(t,c)}^2} < R_{m+1}, & m \in [0, M-1) \\
\\
n = floor(\dfrac{\beta \cdot N}{360^\circ}) = floor(\dfrac{c \cdot \alpha \cdot N}{360^\circ}), & n \in [0, N)
\end{cases}
\tag{4}
$$

where $R_m$ is the radius of the $m - th$ circle of $E$, and $R_0 = 0$ is the center of the LiDAR. $floor(*)$ represents the round-down function. The lowest height value of the points in the $grid_{(m,n)}$ is regarded as the ground surface height of this cell, named $E_{(m,n)}$. The points higher than $E_{(m,n)} + Th_g$ in the $grid_{(m,n)}$ are classified as the obstacle; otherwise, they are classified as the ground. $Th_g$ is a ground height threshold, usually assigned as 0.2 m.

Although it is competent for the ground segmentation task under ideal road conditions, the simple ring-shaped elevation map is easily affected by sloped roads and enormous obstacles in realistic scenes. On the one hand, it is difficult to select a suitable value of ring spacing (we call it $\Delta R$) because of the variable ground height on steep roads. On the other hand, it cannot guarantee that there are actual ground points in the $grid_{(m,n)}$ once the point cloud is sparse. For example, as shown in Figure 4a, the ground points on the slope are regarded as obstacles because the $\Delta R$ is too long. By contrast, the sparse obstacle points on the car's roof will be divided into the ground if the $\Delta R$ is short, as shown in Figure 4b.

The ring-shaped elevation conjunction map (RECM) is proposed to reduce these mistakes by extracting the ground height feature. The road surface within a short distance is relatively flat, so there is no sudden change in the road height. We consequently select a small value of $\Delta R$ to avoid under-segmentation (Figure 4c).

Furthermore, we establish a gradient connection between the front and rear grids, linking these isolated areas that can significantly avoid over-segmentation. Concentrating on the grids with steep gradients and use the slope $k$ as the condition for judging the authenticity of its height.

$$k = \arctan\left(\frac{\Delta E}{\Delta R}\right) = \arctan\left(\frac{E_{(m,n)} - E_{(m-1,n)}}{R_m - R_{m-1}}\right) \quad (5)$$

We consider that there are no actual ground points in the $grid_{(m,n)}$ once the $k$ at the $grid_{(m,n)}$ is greater than the road's maximum slope $\delta_{max}$, and then correct the $E_{(m,n)}$.

$$E_{(m,n)} = \begin{cases} E_{(m,n)}, & k \leq \delta_{max} \\ E_{(m-1,n)} + \Delta R \cdot \tan\delta_{max}, & k > \delta_{max} \end{cases} \quad (6)$$

Figure 4b shows a typical automobile meeting scene where the car on the opposite side occupies an entire cell of elevation map, resulting in no ground points in the $grid_{(m,n)}$. In the previous algorithm, $E_{(m,n)-F}$ is regarded as the ground height in this area, which is the primary cause of over-segmentation. In contrast, the wrong ground height $E_{(m,n)-F}$ is corrected to $E_{(m,n)-T}$ through our proposal because a gradient mutation $\Delta E$ is generated here, as shown in Figure 4d. Thus, the accuracy of pre-classification is increased, even if the $E_{(m,n)-T}$ may not be actual ground height, which is beneficial to our delicate segmentation task.

Algorithm 1 shows the pseudocode for pre-classification through RECM, as long as the input point cloud is traversed twice, the ground can be roughly segmented. All points' label are set to "ground" at first, as shown in line 5. Line 7 indicates that the lowest height of the selected cell is treated as the ground height. Line 9 indicates that the cell height is corrected according to the gradient conjunction. Lines 12 and 13 suggest that points higher than the cell height by a threshold or more are set as obstacles.

---

**Algorithm 1** Coarse segmentation (RECM)

---

**Input:** Raw Point Cloud
**Output:** First Labeled Point Cloud

1  $Th_g \leftarrow$ threshold of the ground height;
2  $\Delta R \leftarrow$ ring spacing of the grid;
3  $\delta_{max} \leftarrow$ road's maximum slope;
4  **for** *each $p_{(t,c)}$ in Cloud* **do**
5       $label_{(t,c)} = ground$;
6       $p_{(t,c)} \in grid_{(m,n)}$;
7       $E_{(m,n)} = \min(E_{(m,n)}, z_{(t,c)})$;
8  **for** *each $E_{(m,n)}$ in E* **do**
9       $E_{(m,n)} = \min(E_{(m,n)}, E_{(m-1,n)} + \Delta R \cdot \tan \delta_{max})$;
10 **for** *each $p_{(t,c)}$ in Cloud* **do**
11      $p_{(t,c)} \in grid_{(m,n)}$;
12      **if** $z_{(t,c)} \geq E_{(m,n)} + Th_g$ **then**
13          $label_{(t,c)} = obstacle$;

---

### 2.3. Jump Convolution Process

A three-channel blank image of $T \times C$ size is established to store the point cloud, and we move origin of the image coordinate system from the upper left to the lower left for the convenience of description. In this way, the position of the point corresponds to the pixels of the image, that is, the pixel $[t, c]$ stores $p_{(t,c)}$. Different labels marked in the first stage correspond to different pixel values. The pixel value of the ground points is $[0, 255, 0]$, the pixel value of the non-ground points is $[255, 0, 0]$. In other words, the green channel stores "ground points", the red channel stores "obstacle points". As shown in Figure 5a, all points are allocated to a corresponding channel after the first step of pre-classification.

Inaccurate segmentation typically occurs at the junctions between the ground and obstacles. That is to say, the category of the points near the red channel ($ch_r$) and the green channel ($ch_g$) is doubtful. The image dilation algorithm is used to describe the credibility of points.

$$ch_r(x, y) = \max_{(x',y'):element(x',y')\neq0} ch_r(x + x', y + y') \tag{7}$$

We dilate the red channel of the original "Ground-Obstacle Image" because of the over-segmentation generated by thresholds. We changed the pixel value of the intersections from $[255, 255, 0]$ to $[0, 0, 255]$ to make the distinction clear. As shown in Figure 5b, the blue channel ($ch_b$) stores "low-confidence points".

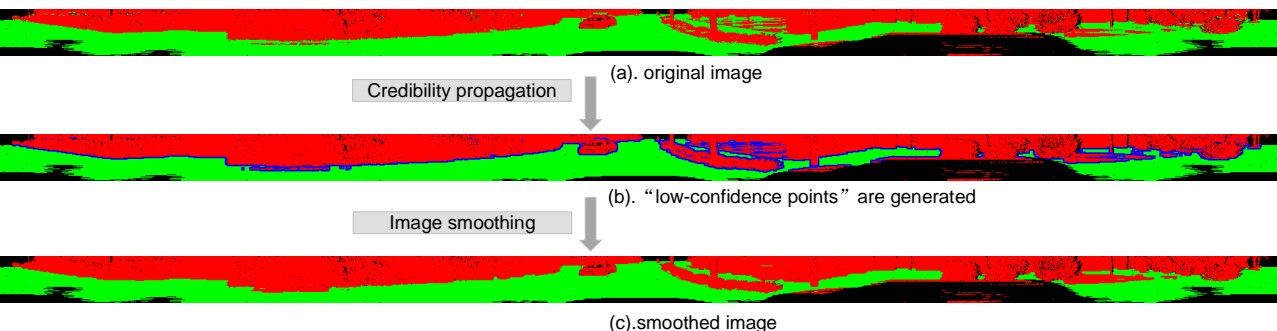

(a). original image

Credibility propagation

(b). "low-confidence points" are generated

Image smoothing

(c). smoothed image

**Figure 5.** Fine segmentation based on jump convolution process (JCP).

Generally, the distance between points belonging to the same object is smaller than the distance between points belonging to different objects. We can determine their final attribution by comparing the distances from "low-confidence points" to different surround-

ing "high-confidence points". As a result, we use $W$ to represent the weight of a pixel and smooth the RGB image depending on the context—a spatial relationship of the neighboring pixel. In the end, the segmentation of the entire image turns into a convolution process. For example, the $5 \times 5$ convolution kernel corresponding to $p_{(t,c)}$ is expressed below.

$$Core_{(t,c)} = \begin{bmatrix} W_{(t+2,c-2)} & W_{(t+2,c-1)} & W_{(t+2,c)} & W_{(t+2,c+1)} & W_{(t+2,c+2)} \\ W_{(t+1,c-2)} & W_{(t+1,c-1)} & W_{(t+1,c)} & W_{(t+1,c+1)} & W_{(t+1,c+2)} \\ W_{(t,c-2)} & W_{(t,c-1)} & 0 & W_{(t,c+1)} & W_{(t,c+2)} \\ W_{(t-1,c-2)} & W_{(t-1,c-1)} & W_{(t-1,c)} & W_{(t-1,c+1)} & W_{(t-1,c+2)} \\ W_{(t-2,c-2)} & W_{(t-2,c-1)} & W_{(t-2,c)} & W_{(t-2,c+1)} & W_{(t-2,c+2)} \end{bmatrix} \tag{8}$$

Among them, $W_{(*)}$ represents the weight of the neighborhood point $p_{(*)}$ and is generated by the Euclidean distance $dis_{xyz}$. The connection between the two points is negatively related to the $dis_{xyz}$, so that we use the $e^{-dis_{xyz}}$ to express the relationship. To be more practical, we also stipulate that the point is irrelevant if $dis_{xyz} > Th_d$, and an amplification factor $s$ is added for data enhancement.

$$D_{(t+i,c+j)} = \begin{cases} 0, & dis_{xyz} > Th_d \quad || \quad (i=0, j=0) \\ \exp(-s \cdot dis_{xyz}), & otherwise \end{cases} \tag{9}$$

Finally, the normalized weighted distance is used as the element of the convolution kernel.

$$W_{(t+i,c+j)} = \frac{D_{(t+i,c+j)}}{\sum\limits_{u=-2,v=-2}^{u=2,v=2} D_{(t+u,c+v)}} \tag{10}$$

We perform a jump convolution operation on the image because we are only interested in "low-confidence points". As shown in Figure 6, when the convolution kernel slides on the image, if it is facing a "high-confidence point", then skip this calculation. If it is facing a "low-confidence point" $p_{(t,c)}$, then update the image by comparing the convolution result of the red channel ($Score_r(t,c)$) and the convolution result of the green channel ($Score_g(t,c)$).

$$Score_r(t,c) = ch_r(t,c) * Core_{(t,c)} = \sum\limits_{i=-2,j=-2}^{i=2,j=2} ch_r(t+i, c+j) \cdot W_{(t+i,c+j)}$$

$$Score_g(t,c) = ch_g(t,c) * Core_{(t,c)} = \sum\limits_{i=-2,j=-2}^{i=2,j=2} ch_g(t+i, c+j) \cdot W_{(t+i,c+j)} \tag{11}$$

It is considered that $p_{(t,c)}$ belongs to an obstacle once $Score_r(t,c) > Score_g(t,c)$, and the pixel value of the image coordinate $[t, c]$ updates to $[255, 0, 0]$; otherwise, the pixel value becomes $[0, 255, 0]$.

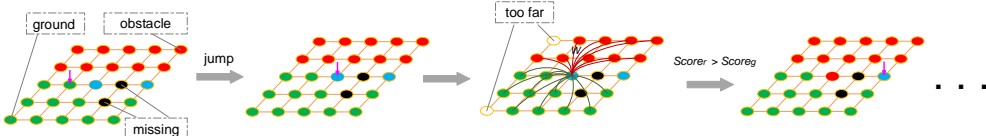

**Figure 6.** Find and re-classify "low-confidence points".

Because the reclassification of blue pixels is an iterative update process, the previous calculation results will affect the subsequent results, so it is necessary to ensure the accuracy of the current calculation. Considering that the nearby point cloud is more "dense" and the distant point cloud is more "sparse", prioritizing the reclassification of nearby low-confidence points will meet our expectations. We choose the direction from the bottom to the top of the

image for convolution to avoid under-segmentation, the final smooth result of the image is shown in the Figure 5c. Similarly, we give the pseudocode of this Algorithm 2.

---

**Algorithm 2** Fine segmentation (JCP)

**Input:** Original Image
**Output:** Final Labeled Point Cloud

1  *Image* $\leftarrow$ the original iamge;
2  $cv :: split(Image, channels)$;
3  $cv :: dilate(channels.at(0), src, (5,5))$;
4  $channels.at(0) = src$;
5  $cv :: merge(channels, Image)$;
6  **if** $Image.at < vec3b > (t,c) == [255, 255, 0]$ **then**
7     $Image.at < vec3b > (t,c) = [0, 0, 255]$;
8  $queue \leftarrow$ sorted pixel coordinates of the blue channel;
9  **while** $!queue.empty()$ **do**
10     $coor_{(t,c)} = queue.front()$;
11     $Score_r(t,c) = Image.at < vec3b > (t,c)[0] * Core_{(t,c)}$;
12     $Score_g(t,c) = Image.at < vec3b > (t,c)[1] * Core_{(t,c)}$;
13     **if** $Score_r(t,c) > Score_g(t,c)$ **then**
14       $Image.at < vec3b > (t,c) = [255, 0, 0]$;
15       $label_{(t,c)} = obstacle$;
16     **else**
17       $Image.at < vec3b > (t,c) = [0, 255, 0]$;
18       $label_{(t,c)} = ground$;
19     $queue.pop()$;

---

Using the *OpenCV* library, the image type is RGB. Lines 2 to 5 indicate that the original image is split based on channels, a $5 \times 5$ kernel dilates the red channel, and the other two channels are merged with the new red channel to generated "low-confidence points". Lines 6 to 8 indicate that the "low-confidence points" are placed in a separate channel and stored in a *queue* data structure in order. Lines 11 and 12 are required to perform a convolution calculation on a specific position of the image. Lines 13 to 17 indicate that the image's pixel value and the corresponding point category are changed according to two possible results.

## 3. Results

This section verifies the flexibility and superiority of the proposed method. We first prove the progressiveness of each module, then compare our method with some previous works [15,23,34] to demonstrate some of our advantages.

### 3.1. Datasets and Evaluation Indicators

We conduct the test based on the Robot Operating System (ROS) on the same computer (Intel I7-8700 CPU and 16 GB RAM) to ensure consistency. Then, we run the algorithms on our self-developed autonomous driving experimental platform (Figure 7) to evaluate its actual working conditions. In all experiments, our experimental parameters are set as follows: $Th_g = 0.2$ m, $\Delta R = 2$ m, $\delta_{max} = 7°$, $Th_d = 1$ m, and $s = 5$, and we select a $5 \times 5$ kernel for image dilation and select 24-pixels neighborhood for smooth.

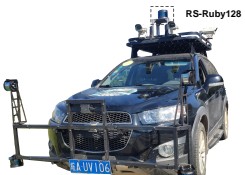

**Figure 7.** Autonomous driving experimental platform: "Intelligent Pioneer No. 3".

Common scenarios simulation tests and extreme scenarios simulation tests are conducted on the Semantickitti dataset, the Koblenz dataset, and our self-made dataset.

We use the labeled sequence (00-10) on the Semantickitti dataset [41]—a total of 23,201 scans data consist of urban traffic, residential areas, highways, and other scenes. It uses the Velodyne HDL-64E LiDAR scanner, which can generate more than 100,000 points per scan. To judge the effectiveness of the method more realistically, we divide the 32 effective categories of Semantickitti into two classes, which are "ground" (including road, sidewalk, and parking) and "non-ground". According to the danger level of obstacles to autonomous driving, we further define some objects in the "non-ground" as "major obstacle" (including human, vehicle, trunk, pole, traffic sign, building, and vegetation). As shown in Figure 8, they are represented by green, red, and fuchsia, respectively. As the "major obstacle" accounts for a substantial proportion in the class "non-ground", it is easy to observe whether there is over-segmentation and judge the safety of the algorithm.

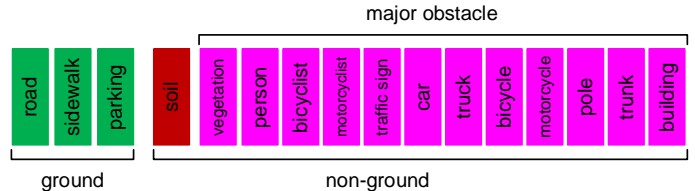

**Figure 8.** Objects included in "ground" and "non-ground".

The Koblenz dataset [43] is also produced through 64-beam LiDAR. Its scenes are more diverse than Semantickitti, including 426 scans for the farm, 644 scans for the campus, and 263 scans for the forest. Similarly, we divide the data into two classes: "ground" and "non-ground". As there is no occlusion from buildings, the point cloud range of this dataset is extensive, which can be used to verify the segmentation effect on different terrains.

Our self-made dataset is collected and produced by our own autonomous driving experimental platform. Its primary LiDAR is an RS-Ruby128 scanner, which collects data at a frequency of 10 Hz, and the output number of points in each scan is twice the Velodyne HDL-64E. We collected and labeled a total of 1148 scans of point cloud data in Hefei, China. It contains 650 scans for city road and 498 scans for field road, and the data are also divided into two classes as above.

The experiment is tested based on the following four aspects:

- the ground segmentation effect of the method;
- the security of the method;
- the time delay of the method; and
- the stability of the method in different environments.

We use $IoU_g$ (intersection-over-union of the "ground"), $Recall_g$ (recall rate of the "ground"), $Recall_{mo}$ (recall rate of the "major obstacle"), and $Delay_t$ (processing time for one scan) as the evaluation indicators of methods' performance. Their calculation formulas are as follows:

$$IoU_g = \frac{1}{n} \sum_{v=1}^{n} \frac{TP_g}{TP_g + FP_g + FN_g} \tag{12}$$

$$Recall_g = \frac{1}{n} \sum_{v=1}^{n} \frac{TP_g}{TP_g + FN_g} \tag{13}$$

$$Reacll_{mo} = \frac{1}{n} \sum_{v=1}^{n} \frac{TP_{mo}}{TP_{mo} + FN_{mo}} \tag{14}$$

$$Delay_t = \frac{Time}{n} \tag{15}$$

where $TP_g$, $FP_g$, and $FN_g$ represent the number of True Positive, False Positive, and False Negative predictions for class "ground", respectively. $TP_{mo}$ and $FN_{mo}$ represent the number of True Positive and False Negative predictions for "major obstacle". *Time* represents the time spent on the program. $n$ Indicates the number of scans. The first two indicators are used to evaluate the accuracy of the segmentation result. In general, if the ground segmentation effect of a specific method is excellent, its corresponding $IoU_g$ and $Recall_g$ will be high. Furthermore, the $Recall_g$ can also reflect the degree of under-segmentation, it must be kept above 95% to ensure smooth driving of the AVs. The $Recall_{mo}$ is used to evaluate the method's safety because it represents the ability to detect critical obstacles, and less than 90% indicates severe over-segmentation. The time indicator is used to evaluate the rapidity. It is considered that the processing time of the ground segmentation for a low-frequency LiDAR (10 Hz) should not exceed 40 ms, for a high-frequency LiDAR ($\geq$20 Hz) should not exceed 20 ms.

### 3.2. Performance Verification of Each Module

As mentioned in Section 2.2, the ring-shaped elevation map (REM) algorithm is not suitable for sparse point clouds, and the obstacle will be incorrectly divided into the ground in complex scenes, resulting in the loss of obstacle information. By adding ground height characteristic in the coarse-segmentation stage, the ring-shaped elevation conjunction map (RECM) can effectively improve the segmentation effect. The jump convolution process (JCP) further enhances the segmentation effect because "low-confidence points" are reclassified in the fine-segmentation stage. We tested these modules on the Semantickitti dataset. Table 1 shows the segmentation results, and Figure 9 shows the overall segmentation effect.

**Table 1.** The influence of each module of the proposed algorithm on performance.

| Method | $IoU_g$ (%) | $Recall_g$ (%) | $Recall_{mo}$ (%) | $Delay_t$ (ms) |
|:---:|:---:|:---:|:---:|:---:|
| REM [1] | 67.73 | **99.40** | 83.65 | **1.99** |
| RECM [2] | 72.20 | 99.20 | 88.63 | 2.00 |
| RECM + JCP [3] | **76.50** | 98.07 | **96.04** | 8.61 |

[1] Ring-shaped elevation map. [2] Ring-shaped elevation conjunction map. [3] Jump convolution process.

It can be seen from Table 1 that the $IoU_g$ of the REM algorithm is very low, and the $Recall_{mo}$ does not even reach 90%. Therefore, classifying the point cloud based only on the relative height difference will be accompanied by obvious over-segmentation. Compared with the REM algorithm, the RECM algorithm increases the calculation time by 0.01 ms, but it improves the segmentation effect and weakens the over-segmentation phenomenon. The result shows that establishing the ground gradient relationship helps reduce the false prediction of the ground points. The subsequent JCP module not only further improves the segmentation effect, but also dramatically increases the security performance. Although the $IoU_g$ dropped by about 1% after adding the JCP module, it remained at a high level. In the end, the $Recall_g$ and the $Recall_{mo}$ reach 98.07% and 96.04%, respectively. It is an impressive effect for traditional methods. In short, the RECM algorithm increases the ground segmentation effect by approximately 4% compared with the most straightforward REM algorithm, and the JCP module increases the segmentation effect by approximately 5% again on the basis of RECM. It is indicating that each module plays an uplifting role in the segmentation.

We select two representative cases in the test scenario, as shown in Figure 9. Each group's left and right pictures can be viewed as a good comparison. In particular, the color description of the benchmark image can be found in Figure 8.

The first group has a few static "major obstacle" and much soil. The contour of the ring-shaped elevation map can be seen very intuitively in the b→1. The REM algorithm only calculates the relative height difference of the internal points of the individual grid, resulting in the presence of ground points in each grid. Therefore, there are many false detections, which is also the main reason that leads to over-segmentation. In the RECM algorithm, the adjacent grids are no longer isolated, and most false detections can be corrected, as shown in c→1. JCP inherited this advantage and smoothed the junction between the ground and non-ground, making the ground look more coherent (d→1). However, when facing some low and flat obstacles, such as the soil in the d→2, it is impossible to filter them out of the ground only by the height and distance characteristics, so our method cannot detect these obstacles.

The second group is a scene where multiple obstacles coexist, the defects of the REM algorithm are exposed again. The occlusion of the front right by dynamic obstacles resulted in the lack of ground points, so the bottom of the car and the building are mistakenly divided into the ground (b→3). In actual driving, the AVs may not take avoidance measures in time because of these missed inspections. It is an extremely dangerous mistake for AVs. Although RECM can make up for some errors because it reconstructs the ground height, the car in the park is still incompletely detected. JCP reclassifies the "low confidence points" on the edge of "non-ground", ideally separate the ground and obstacle (d→3), and dramatically improves the detection distance of obstacles (d→4).

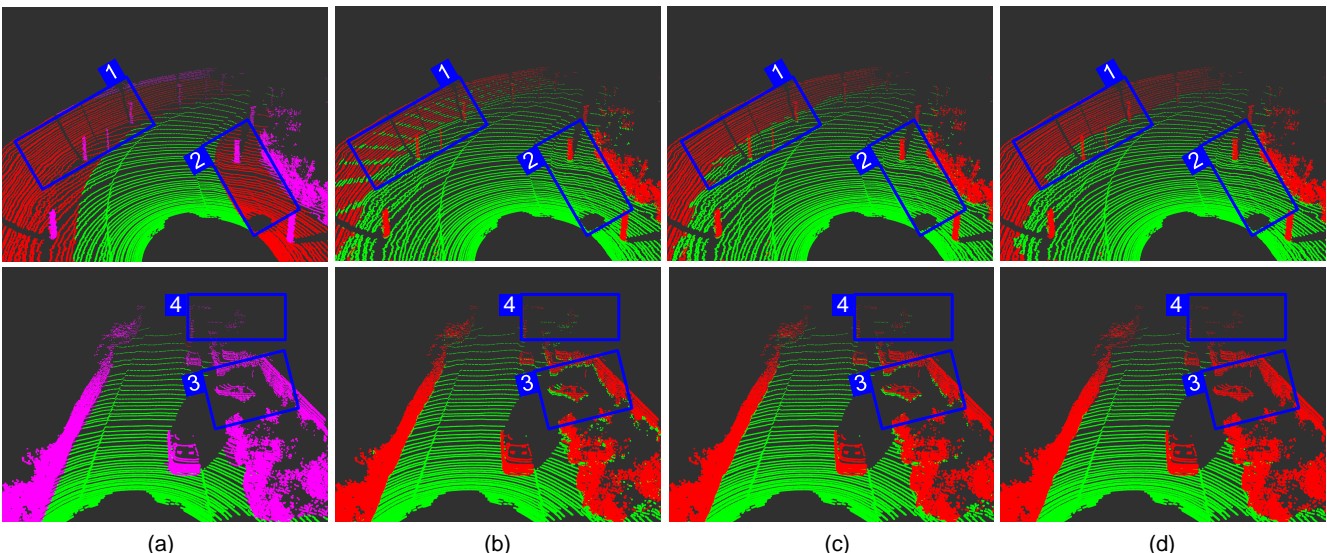

      (a)                 (b)                 (c)                 (d)

**Figure 9.** Segmentation result of each module in different scenes. Grounp 1 is the ordinary road scene; Grounp 2 is the residential road scene. (**a**) Benchmark; (**b**) REM; (**c**) RECM; (**d**) RECM + JCP.

### 3.3. Comparative Experiment with Other Methods

We take the works in [15,23,34] as our experimental references. They are representative studies in this field and have shown promising results: the work in [15] first segments the point cloud by the local angle and height features of the front and back points, and then improves the segmentation effect through the horizontal clustering. The work in [23] establishes a LiDAR depth image, and then quickly filters out ground points based on the angle formed by the LiDAR center and the next point on the depth image. The work in [34] initializes the Markov random field (MRF) with the seed points obtained in advance. Then, the authors use the max-flow/min-cut algorithm to cut the deep image of the point cloud. We test all methods on the three datasets to analyze their segmentation effect, security, time delay, and stability. Because some algorithms in the references are susceptible to parameters, we use the best parameters for different datasets instead of the parameters given in their articles.

### 3.3.1. Structured Road Test

We selected 00-10 sequences from the Semantickitti dataset for this test. Semantickitti is a typical traffic dataset, which includes not only daily urban traffic but also residential traffic.

Table 2 lists the specific segmentation results on each sequence. Obviously, the method in [23] performs the worst in all methods, and the segmentation evaluation indicators are the lowest. In addition, the average value of $Recall_{mo}$ is only 87.58%, indicating that there will be apparent over-segmentation when using this algorithm. In contrast, the other methods obtain relatively good segmentation results. The method in [15] has the highest $Recall_g$ and the lowest $Delay_t$, indicating that it is feasible to segment the point cloud through the joint features of the angle and height. The $Recall_g$ of the proposed method ranks second while maintaining a low time delay. Compared with the method in [15], our $Recall_g$ has dropped by less than 0.6%, but our average $IoU_g$ has increased by 3%, indicating that there are fewer False Positive predictions for class "ground". At the same time, our method has the highest $Recall_{mo}$, with a value of 96%, which proves that our method has the most reliable security guarantee. It is worth mentioning that, benefiting from using global depth features instead of local features, the segmentation indicators of the method in [34] are relatively close to our method. Nevertheless, its time delay is about three times ours, so it is unsuitable for some high-frequency LiDAR.

**Table 2.** The evaluation results on the Semantickitti dataset.

| Method | Sequences Scenes | 00 Residential Area | 01 Highway Scene | 02 City Traffic | 03 Residential Area | 04 City Traffic | 05 Residential Area | 06 City Traffic | 07 Residential Area | 08 Residential Area | 09 City Traffic | 10 City Traffic | Mean |
|---|---|---|---|---|---|---|---|---|---|---|---|---|---|
| | | | | | | $IoU_g$(%) | | | | | | | |
| Chu [15] | | 79.96 | 70.63 | 83.84 | 64.53 | 76.89 | 76.63 | **54.84** | 81.88 | 68.68 | 77.98 | 72.59 | 73.49 |
| Bogoslavskyi [23] | | 73.81 | 67.28 | 79.26 | 61.12 | 73.50 | 74.81 | 53.27 | 76.00 | 76.24 | 74.17 | 67.68 | 69.56 |
| Huang [34] | | 82.47 | 72.58 | 85.72 | 67.46 | 78.29 | 80.00 | 54.42 | **84.80** | 71.49 | 79.82 | 75.06 | 75.65 |
| Our | | **82.92** | **74.69** | **86.32** | **67.71** | **79.69** | **80.76** | 54.83 | 84.56 | **72.07** | **80.88** | **77.12** | **76.50** |
| | | | | | | $Recall_g$(%) | | | | | | | |
| Chu [15] | | **99.17** | **99.59** | **98.37** | 98.17 | **99.71** | **98.80** | **97.53** | **98.99** | **98.92** | **98.76** | **96.66** | **98.61** |
| Bogoslavskyi [23] | | 96.29 | 96.25 | 96.36 | 96.73 | 98.54 | 95.14 | 93.89 | 96.34 | 96.55 | 96.96 | 95.35 | 96.22 |
| Huang [34] | | 98.94 | 97.84 | 97.48 | 98.27 | 99.43 | 98.41 | 95.93 | 98.69 | 98.76 | 97.15 | 95.05 | 97.81 |
| Our | | 98.80 | 99.11 | 98.12 | **98.63** | 99.29 | 98.44 | 96.00 | 98.66 | 98.78 | 97.80 | 95.11 | 98.07 |
| | | | | | | $Recall_{mo}$(%) | | | | | | | |
| Chu [15] | | 94.25 | 92.47 | 90.70 | 88.61 | 92.82 | 91.28 | 95.23 | 93.53 | 94.08 | 92.65 | 90.56 | 92.38 |
| Bogoslavskyi [23] | | 89.83 | 73.23 | 88.53 | 89.26 | 80.94 | 91.35 | 93.00 | 89.40 | 89.14 | 90.86 | 87.79 | 87.58 |
| Huang [34] | | 95.60 | 95.50 | 93.38 | 92.42 | 94.78 | 93.76 | 95.97 | 95.30 | 95.44 | 94.63 | 93.32 | 94.55 |
| Our | | **96.92** | **97.46** | **94.12** | **93.12** | **97.52** | **94.45** | **98.44** | **95.60** | **97.26** | **96.39** | **95.12** | **96.04** |
| | | | | | | $Delay_t$(ms) | | | | | | | |
| Chu [15] | | **6.41** | 11.02 | **7.02** | **7.66** | **7.42** | **6.79** | **7.46** | **6.54** | **6.60** | **6.91** | **6.11** | **7.27** |
| Bogoslavskyi [23] | | 10.23 | **8.60** | 10.52 | 10.44 | 10.51 | 10.06 | 10.15 | 10.09 | 10.15 | 10.51 | 10.28 | 10.12 |
| Huang [34] | | 24.32 | 23.37 | 25.42 | 25.79 | 25.54 | 25.57 | 25.80 | 24.16 | 25.13 | 25.57 | 24.77 | 25.04 |
| Our | | 8.09 | 8.97 | 7.90 | 9.59 | 8.78 | 8.71 | 8.92 | 7.81 | 8.51 | 8.80 | 8.61 | 8.61 |

Figure 10 captures the segmentation effect of some scenes. Group 1 is the scene of the vehicle speeding on the highway, and we mark out the areas that are worthy of attention. We can see that all methods can completely segment the car from the point cloud for scenes with a small number of obstacles, as shown in box 1. However, [23,34] both have serious segmentation problems in other regions. The method in [23] hardly detects obstacles with gradual changes in depth. As shown in c→2, almost the entire railing is regarded as the ground, and this over-segmentation phenomenon is undoubtedly fatal for Avs driving at high speeds. On the contrary, the method in [34] relies on the histogram to determine probability in the graph cut algorithm, which will cause serious under-segmentation. As shown in d→3, the entire road ahead is viewed as an obstacle. This error will cause the AVs to make brake suddenly, which is very dangerous for high-speed road sections. In contrast, our method and that in [15] have the expected segmentation effect in this scene.

Group 2 is the scene where the experimental car following a car on a residential road. Except for our method, the rest of the methods have problems in detecting the ground.

The angle and depth features are invalid because the car hood is relatively flat, so the methods in [15,23] both regard it as the ground (b→4, c→4). The method in [34] also fails (d→4) because the ordinary elevation map considers the lowest point in the grid as the ground, but the lowest point in this area happens to be on the hood. Our method can correct the wrong ground height in this area to achieve the best segmentation effect (e→4). We also noticed that the method in [15] is particularly sensitive to occlusion. Observe the position in b→5, the building is considered as the ground because of the occlusion generated by the front right car. Meanwhile, the segmentation of obstacles in [23] appears intermittent. The method in [34] has defects in the connection processing between obstacle points and ground points. As shown in d→6, it often produces "noise points" between obstacles and the ground. In general, our method has the perfect segmentation results and the slightest error detection in the test. For details, please zoom in and observe Figure 10.

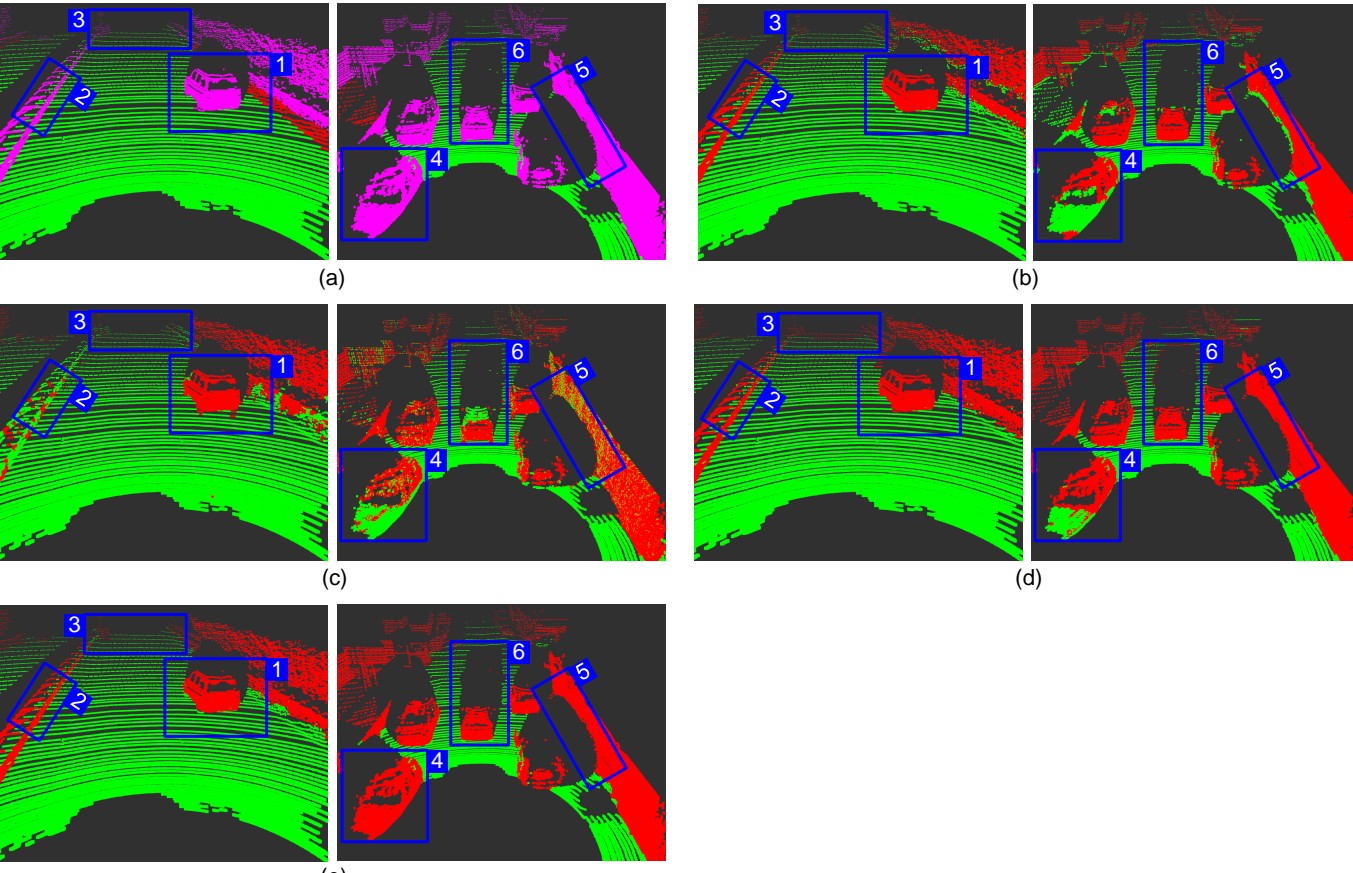

**Figure 10.** The ground segmentation effect of different methods on Semantickitti dataset. (**a**) Benchmark; (**b**) Chu [15]; (**c**) Bogoslavskyi [23]; (**d**) Huang [34]; (**e**) Ours. (It is recommended to zoom in to see the details).

Even though our strategy has a better segmentation effect than the other three methods on this dataset, the test of urban roads alone is not enough. In order to demonstrate the performance of our method convincingly, we next adopt a more rigorous test scenario.

### 3.3.2. Unstructured Road Test

Compared with urban traffic, the point cloud for unstructured roads is more irregular, and the road conditions are bumpier. We choose the FarmMap and ForestMap in the Koblenz dataset as the test scenarios for the first stage, then select the more challenging FieldMap in the self-made dataset as the supplementary test scenario for the second stage. Tables 3 and 4, respectively, show the test results.

**Table 3.** The evaluation results on the Koblenz dataset.

| Method | Environment | FarmMap | ForestMap | Mean |
|---|---|---|---|---|
| $IoU_g(\%)$ | | | | |
| Chu [15] | | 29.00 | 48.92 | 38.96 |
| Bogoslavskyi [23] | | 26.47 | 28.10 | 27.28 |
| Huang [34] | | 33.00 | 57.91 | 45.46 |
| Our | | **34.97** | **64.57** | **49.77** |
| $Recall_g(\%)$ | | | | |
| Chu [15] | | 96.44 | 98.18 | 97.31 |
| Bogoslavskyi [23] | | 95.17 | 95.02 | 95.10 |
| Huang [34] | | 98.38 | 98.25 | 98.32 |
| Our | | **99.32** | **98.89** | **99.11** |
| $Recall_{mo}(\%)$ | | | | |
| Chu [15] | | 70.28 | 80.56 | 75.42 |
| Bogoslavskyi [23] | | 57.51 | 42.76 | 50.13 |
| Huang [34] | | 78.11 | 88.25 | 83.13 |
| Our | | **83.91** | **91.97** | **87.94** |
| $Delay_t(\%)$ | | | | |
| Chu [15] | | **9.56** | **6.43** | **8.00** |
| Bogoslavskyi [23] | | 10.33 | 10.43 | 10.38 |
| Huang [34] | | 26.86 | 24.43 | 25.65 |
| Our | | 9.23 | 8.55 | 8.89 |

**Table 4.** The evaluation results on the self-made dataset.

| Method | Environment | FieldMap |
|---|---|---|
| $IoU_g(\%)$ | | |
| Chu [15] | | 17.92 |
| Bogoslavskyi [23] | | 15.53 |
| Huang [34] | | 15.72 |
| Our | | **18.95** |
| $Recall_g(\%)$ | | |
| Chu [15] | | 93.29 |
| Bogoslavskyi [23] | | 93.24 |
| Huang [34] | | **97.88** |
| Our | | 97.30 |
| $Recall_{mo}(\%)$ | | |
| Chu [15] | | 82.92 |
| Bogoslavskyi [23] | | 80.14 |
| Huang [34] | | 78.95 |
| Our | | **82.94** |
| $Delay_t(\%)$ | | |
| Chu [15] | | **11.16** |
| Bogoslavskyi [23] | | 16.87 |
| Huang [34] | | 47.51 |
| Our | | 15.62 |

In the first phase of the test, the segmentation indicators of all methods dropped significantly, especially in the farm environment. Because the soil occupies most of the point cloud data in the farm scene, these methods in the experiment have almost no filtering ability on the soil, so the $IoU_g$ becomes very low. The method in [23] not only has the lowest $Recall_g$ and $IoU_g$, but also the $Recall_{mo}$ has just reached 50%, which means that this method is not suitable for the unstructured road at all. The method in [15] is slightly more robust; it has better segmentation performance and improves $Recall_{mo}$ to 75%, which provides higher security compared to [23], indicating that the angle feature has higher stability than the depth feature. The method in [34] and our method are undoubtedly have the most robust anti-interference ability because the $IoU_g$ indicator and the $Recall_{mo}$ indicator are much higher than the other two methods. Like the structured road, although the segmentation accuracy of [34] is close to ours, we have higher security and lower latency (Appendix A.1 shows the stability during continuous operation).

In the second phase, all methods become unstable due to the ups and downs of the ground. The method in [34] has the highest $Recall_g$, but the $IoU_g$ is very low, which is a typical over-segmentation. The method in [15] has the highest $IoU_g$, but the $Recall_g$ is very low, indicating that there are apparent false detections in the ground predictions. Although our method comprehensively achieves the best segmentation effect, the $IoU_g$ is only 18.95%, meaning that there is also over-segmentation. In general, our method performs better than other methods on unstructured roads. Combined with the segmentation effect on structured roads, our method has the strongest robust performance.

Figure 11 shows the segmentation effect in an unstructured road environment. Because the vehicle cannot drive smoothly under such road conditions, the point cloud becomes irregular. The first group is a scene of a sharp turn in the forest scene. It can be seen from the point cloud that the ground is tilted, which makes it very unreliable to extract the angle feature from the point cloud. Therefore, the method in [15] has false detection at the ground tilt (b→1). The method in [23] becomes very unstable under these road conditions. There is much wrong segmentation around, intermittent false detection points appear on the ground, the detection of plants on the left is incomplete, and even the shrubs on the right are regarded as ground (c→2). The method in [34] and our method show reliable segmentation performance, the ground and obstacles around the AVs are perfectly divided, but [34] once again make mistake for the front road (d→2).

The second group is a field scene. Facing this irregular intersection, the method in [15] and that in [23] are helpless, and neither the ground on the left nor the right is detected (b→3, c→3). Although the method in [34] detects the ground on the right, it regards the shadow on the left as an obstacle (d→3). Our result is a qualified ground segmentation. It does not produce false detections on the ground and detects the main boundary on both sides. This way, it can ensure the AVs drive logically on bumpy roads and provide usable boundary information.

Although the segmentation effect of our method on the unstructured roads is not as good as that on the structured roads, our method still shows solid environmental adaptability, indicating that the proposed can be migrated to different terrain environments (the test videos of our method on each dataset can be found in Appendix A.2). Whether it is a sloping ground or a bumpy ground, we rarely detect the road surface as an obstacle, and our method can always catch the most "major obstacle", ensuring accuracy and safety at the same time.

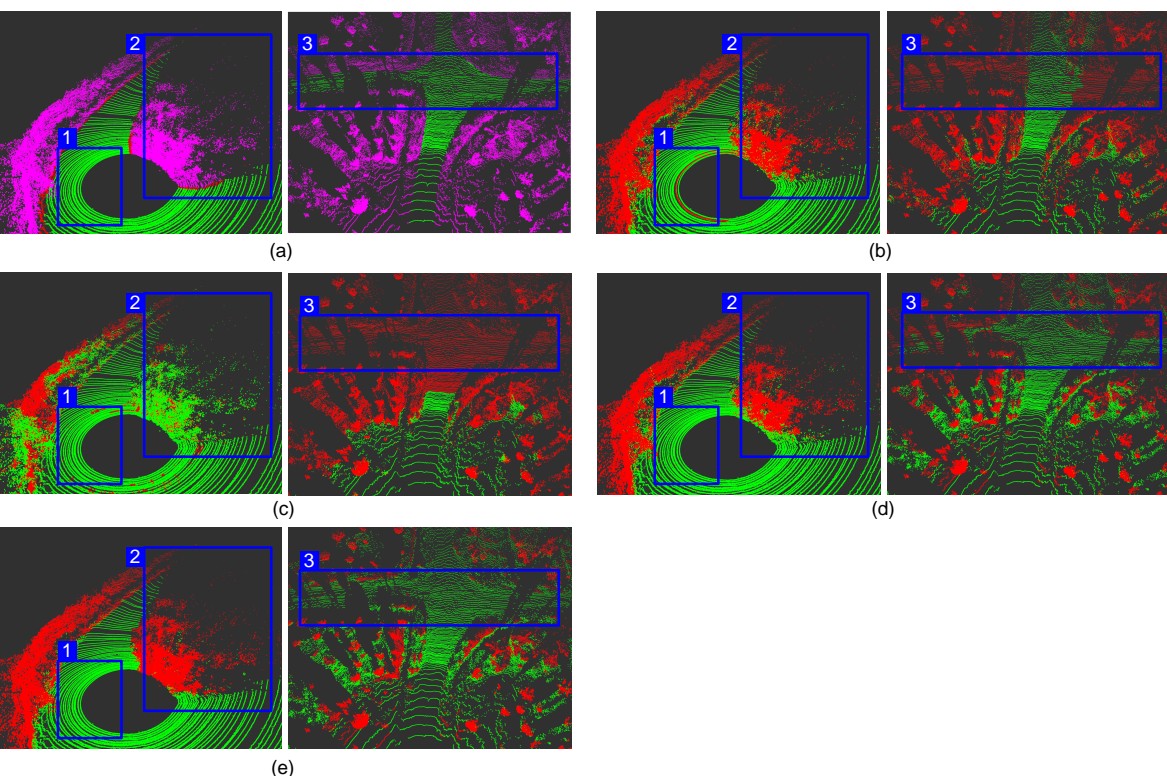

**Figure 11.** The ground segmentation effect of different methods on unstructured road. (**a**) Benchmark; (**b**) Chu [15]; (**c**) Bogoslavskyi [23]; (**d**) Huang [34]; (**e**) Ours. (It is recommended to zoom in to see the details.)

## 4. Discussion

The above experiments prove that the method proposed in this paper can obtain accurate segmentation results in various scenes and run in real-time. Compared with the simple elevation map method, our advantage is that we can correct the wrong ground height through the conjunction between the grids. On the one hand, the corrected ground height reduces the serious over-segmentation in the coarse segmentation stage. On the other hand, it generates more high-confidence points used to re-classify the low-confidence points in the fine segmentation stage. We mainly use height difference and distance features, which are stable even if the point cloud is rotated. Our method is more sensitive to obstacle detection and robust than methods based on local point cloud features, such as angle or depth. It generates fewer False Positive predictions for "ground" even if the ground is inclined or the vehicle is bumpy. On rugged roads, the Euler angles of the car change constantly, and the angle characteristics and depth characteristics become irregular. Therefore, the segmentation performance of the method in [15] and that in [23] on the unstructured road drops rapidly. Compared with the MRF conditional random field method, our method has better real-time performance because optimizing the energy function in MRF is a time- and memory-consuming process. In addition, there are significant differences between global and local features in unstructured roads, so our segmentation effect is better than method [34] in the face of irregular point clouds.

Although our method can achieve a good segmentation effect in most cases, we also found that there are several limitations: (1) The coarse segmentation stage only uses the height characteristics of the point cloud, making almost all soils with the same height as the road are divided into the class "ground", which is very obvious in the open field experiments. (2) Our method cannot solve or even deepen the under-segmentation situation because the application premise of our fine segmentation is that the result of coarse segmentation is over-segmented. (3) The two modules in our method use the same thread, which cannot give full play to the performance of the hardware. Using dual threads can further reduce the time delay.

## 5. Conclusions

Based on previous work, this paper divides the segmentation of the LiDAR point cloud into two stages—from coarse to fine. The first stage uses improved local point cloud features and ground features to quickly classify the original point cloud. The ring-shaped conjunction elevation map (RECM) algorithm has excellent segmentation capabilities for dense point clouds. As the establishment of the gradient connection corrects the wrong ground height, it also has a good detection ability for non-ground points in sparse point clouds. In the second stage, we map the point cloud to the image, use the image dilation algorithm to generate the "low-confidence points", and finally use the jump convolution process (JCP) to re-classify their categories and smooth global points. We compared the other three methods on the three datasets, including urban and field driving environments. The results show that the proposed method has a more accurate segmentation effect, a more reliable safety, a more efficient performance, and a more robust environmental adaptability than others.

We noticed in our experiments that it is challenging to distinguish the road surface and the soil only by the height and distance information of the point cloud, and relying on the angle relationship of the point cloud often leads to incorrect predictions. Therefore, finding some deep features suitable for irregular point clouds will be our next research direction. In addition, we noticed that the application of convolutional neural networks (CNN) is gradually expanding. However, deep learning directly on the 3D point cloud requires good hardware support, and the anti-interference performance of the convolutional neural network is abysmal, so we think that applying the convolutional neural network to the 2.5D depth image projected by LiDAR may become an auxiliary method for segmenting point clouds.

**Author Contributions:** Conceptualization, Z.S. and L.L.; methodology, Z.S. and W.H.; formal analysis, J.Y.; data curation, W.H.; writing—original draft preparation, Z.S.; writing—review and editing, L.L., Z.W. and H.L.; visualization, Z.S.; supervision, L.L.; funding acquisition, H.L. All authors have read and agreed to the published version of the manuscript.

**Funding:** This work was supported by National Key Research and Development Program of China (Nos. 2020AAA0108103, 2016YFD0701401, 2017YFD0700303, and 2018YFD0700602), Youth Innovation Promotion Association of the Chinese Academy of Sciences (Grant No. 2017488), Key Supported Project in the Thirteenth Five-year Plan of Hefei Institutes of Physical Science, Chinese Academy of Sciences (Grant No. KP-2019-16), Natural Science Foundation of Anhui Province (Grant No. 1808085QF213) and Technological Innovation Project for New Energy and Intelligent Networked Automobile Industry of Anhui Province.

**Institutional Review Board Statement:** Not applicable.

**Informed Consent Statement:** Not applicable.

**Data Availability Statement:** Publicly available datasets were analyzed in this study. These data can be found here: http://www.semantic-kitti.org/ (accessed on 1 August 2021), http://robots.uni-koblenz.de/datasets (accessed on 1 August 2021). Some data presented in this study are available on request from the corresponding author. The data are not publicly available due to the confidentiality clause of some projects.

**Conflicts of Interest:** The authors declare no conflict of interest.

## Appendix A

*Appendix A.1. Stability of Running Time*

We intercept 1000 consecutive scans on the self-developed autonomous driving platform to test the stability of our method. Figure A1 shows the processing time of our program. Although the curve fluctuates, the peak value of 128-beam LiDAR does not exceed 20 ms, indicating that our method can be applied to actual scenarios.

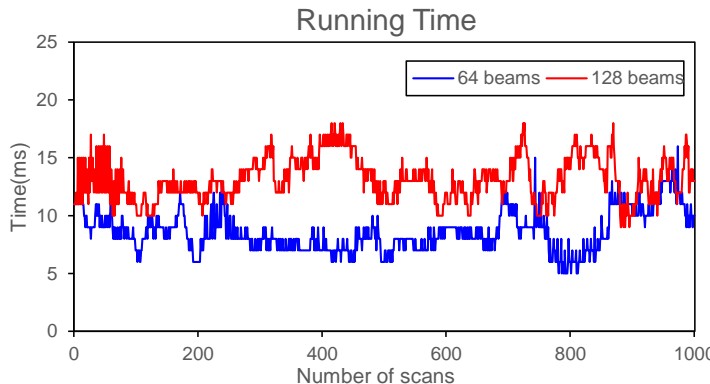

**Figure A1.** The running time of our method on 1000 consecutive scans.

*Appendix A.2. Experiment Videos*

Three video clips of experimental tests are available on YouTube.
Urban test: https://youtu.be/mV3TtKd1UGw (accessed on 1 August 2021).
Forest test: https://youtu.be/2X9P2Zj3AIQ (accessed on 1 August 2021).
Field test: https://youtu.be/uy4Wf3W-4_g (accessed on 1 August 2021).
These webs can be accessed permanently.

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
