# Peer review of "Fast Ground Segmentation for 3D LiDAR Point Cloud Based on Jump-Convolution-Process"

_remotesensing, doi:10.3390/rs13163239_

Round 1
Reviewer 1 Report
I would like to congratulate the authors for the presented work. The authors present a framework for accurately segmenting the ground plane mainly for autonomous driving scenario. The paper is well presented and is technically sound.
Ground segmentation is a well known research problem with several already existing techniques. The authors have provided a thorough state of the art review and also given a clear motivation for using their method.
The experimental validation is also well done, several experiments have been performed on standard dataset and also on the custom datasets, comparing with 2 other state of the art methods. The results do show the proposed method is not only accurate but also computational faster compared to other methods.
I accept the paper for publication in the journal.
Author Response
Dear reviewer:
Thank you very much for taking the time to read my manuscript and make detailed comments during your busy schedule. I am very grateful for your affirmation of my work. Your reply has increased my confidence in my future research work.
Best wishes to you.
Kind regards,
Zhihao Shen
Reviewer 2 Report
Review comment and suggestions are listed in the PDF.

Author Response

(The authors gave the same response as above.)

Reviewer 3 Report
This manuscript is interesting because it emphasizes the need of LiDAR data for self-driving use in autonomous vehicles (AVs). This is to ensure driving safety can be achieved for AVs. However, this manuscript does not explain in detail the need for AVs to achieve a better level of safety. It should be discussed first to find out the needs or problems that involves with AVs. And this also needs to be discussed again in the findings and discussion section. However, this relevant matter is only being stated at the beginning of the manuscript.
I believe there is a possibility that JCP can help AVs problems. Since this manuscript does not state specifically what the problems of AVs are, it is quite difficult to identify what is to be resolved. In the abstract of the study, it is stated that JCP solves the problem of unavailability on complex terrains, excessive time and memory usage, and additional pre-training requirements. However, this problem is more towards the problems faced by the existing algorithms so far and not the problems of AVs. Plus, the proposed solutions and findings are not discussed systematically and not re-linked to AVs issues.
This manuscript needs to be rewrite in a more organized writing structure. For example, placing some of the results of the study in the Introduction section can be somehow confusing for the reader. Furthermore, this manuscript is lacking in the discussion related to the results of the study. Usually, there is a dedicated section (i.e., Discussion) to explain in detail about the findings made. Unfortunately, that section is not available in this manuscript. In addition, Section 3.5 is also seen as too short to be as a section (one short paragraph with a figure). Not sure if it should be placed as one section or included in other sections. Therefore, the writing of this manuscript is suggested to be reviewed again to improve its quality.
The use of English in this manuscript is recommended for revision to avoid typographical errors. Maybe English is not the writer's native language, but maybe the writer can use simple English and keep it short for each point that the author wishes to address. This will make it easier for the reader to understand what is being conveyed.
Author Response

(The authors gave the same response as above.)

Round 2
Reviewer 3 Report
This manuscript has addressed the comments given in the first-round review process. The quality of the manuscript improved, and it makes the manuscript interesting for the readers compared with the previous version.